# Evaluation of Fecal Calprotectin, Serum C-Reactive Protein, Erythrocyte Sedimentation Rate, Seromucoid and Procalcitonin in the Diagnostics and Monitoring of Crohn’s Disease in Children

**DOI:** 10.3390/jcm11206086

**Published:** 2022-10-15

**Authors:** Katarzyna Akutko, Barbara Iwańczak

**Affiliations:** 2nd Department of Paediatrics, Gastroenterology and Nutrition, Faculty of Medicine, Wroclaw Medical University, 50-369 Wroclaw, Poland

**Keywords:** biomarkers, non-invasive, activity, diagnosis, monitoring

## Abstract

Background: The development of diagnostic and monitoring algorithms for Crohn’s disease based on non-invasive methods is of particular importance in children and is the subject of many studies. Objectives: Evaluate the usefulness of fecal calprotectin, serum C-reactive protein, erythrocyte sedimentation rate, seromucoid and procalcitonin in the differential diagnosis of non-inflammatory gastrointestinal tract diseases and Crohn’s disease in children and their usefulness in determining the phenotype of Crohn’s disease. Material and methods: Forty-seven children with non-inflammatory gastrointestinal tract diseases and fifty-four with Crohn’s disease were enrolled. Clinical and endoscopic activity was evaluated based on the Pediatric Crohn’s Disease Activity Index (PCDAI) and the Simple Endoscopic Score for Crohn’s Disease (SES-CD). Results: Fecal calprotectin, C-reactive protein, erythrocyte sedimentation rate and seromucoid were significantly higher in children with Crohn’s disease than in controls (*p* < 0.001). Fecal calprotectin correlated with clinical and endoscopic activity according to the Pediatric Crohn’s Disease Activity Index (r = 0.338; *p* = 0.012) and the Simple Endoscopic Score for Crohn’s Disease (r = 0.428; *p* = 0.001). Non-invasive biomarkers did not correlate with the location and clinical manifestation of Crohn’s disease. Conclusions: Fecal calprotectin, C-reactive protein, erythrocyte sedimentation rate and seromucoid are useful in the differentiation of Crohn’s disease from non-inflammatory gastrointestinal tract diseases in children and in monitoring the clinical course of Crohn’s disease, but not in evaluating activity and phenotype of the disease.

## 1. Introduction

Crohn’s disease (CD) in children often presents insidiously, without typical clinical symptoms such as abdominal pain, diarrhea and weight loss. This may delay the diagnosis for a few years [1,2,3]. The course of CD is often more aggressive, and the development of complications (i.e., fistulas, intra-abdominal abscesses) is more often observed in children than adults. Children can also develop complications that do not affect adults, i.e., puberty delay, growth retardation, poor peer relationships and the resulting emotional disturbances [1,4]. Delay in diagnosis and initiation of treatment can lead to uncontrolled progress of the disease and evolution from non-stricturing and non-penetrating CD, which is most commonly in children, into stenosing or penetrating disease [5]. Thus, there is an unquestioning need to develop specific, non-invasive and easy-to-carry-out laboratory tests for quick diagnosis and monitoring of the effectiveness of treatment in children with CD. Currently, the choice of therapy depends on the CD phenotype, which is determined based on invasive endoscopic examinations of the gastrointestinal tract. In further years, gastrointestinal endoscopy must be frequently repeated to evaluate the effectiveness and individualization of treatment [2,4]. Chronic abdominal pain is one of the most common causes of pediatric gastroenterology consultations. Therefore, there is a need to use non-invasive tests to identify children with a strong suspicion of organic gastrointestinal tract diseases [6]. An inflammatory biomarker already used in everyday practice is calprotectin, of which the concentration is measured in feces [7,8]. Other selected serum markers are also used to assess CD clinical activity: C-reactive protein, erythrocyte sedimentation rate or seromucoid [9].

## 2. Objectives

The aim of the study was to evaluate the usefulness of non-invasive inflammatory markers: fecal calprotectin (FC), serum C-reactive protein (CRP), erythrocyte sedimentation rate (ESR), seromucoid (AAG) and procalcitonin (PCT) in the differential diagnosis of CD and non-inflammatory gastrointestinal tract diseases in children. The objective of the study was also to assess the utility of these biomarkers for evaluation of clinical activity according to the Pediatric Crohn’s Disease Activity Index (PCDAI) and endoscopic activity of CD in children according to Simple Endoscopic Score for Crohn’s Disease (SES-CD).

## 3. Material and Methods

A total of 101 patients aged 5–18 years hospitalized in the 2nd Department of Pediatrics, Gastroenterology and Nutrition from 2016 to 2017 were included in this prospective cross-sectional study. The study group consisted of 54 patients with CD (30 boys, 55.56% and 24 girls, 44.44%). The study group included patients with CD who were hospitalized both for diagnosis and for the purpose of performing routine tests in patients with previously diagnosed disease, intensification of treatment or exacerbation of inflammatory bowel disease. The control group comprised 47 patients (30 boys, 63.83% and 17 girls, 36.17%) with non-inflammatory gastrointestinal tract diseases: 16/47 (34.04%) with functional abdominal pain, 12/47 (currenl. 53%) with lactose intolerance, 9/47 (19.15%) with functional constipation, 6/47 (12.77%) with irritable bowel syndrome and 4/47 (8.51%) with other inorganic diseases of the digestive tract (i.e., hypercholesterolemia). There was a need to perform gastrointestinal endoscopy in 17 patients (36.17%) of the control group. The age, weight and growth of children in the study and control group were not statistically different.

The diagnosis of CD was based on clinical symptoms, laboratory tests and histological, radiological and endoscopic findings according to the Porto Criteria recommended by The European Society for Pediatric Gastroenterology Hepatology and Nutrition (ESPGHAN). Some patients were diagnosed with CD before enrolling in the study. Endoscopy of the upper and lower gastrointestinal tract was performed in all the patients of the study group. Additionally, radiological diagnostics were performed: all children with CD were examined with an abdomen ultrasound. In total, 80% of those patients also underwent magnetic resonance enterography. CD phenotype was classified according to the Paris Classification [10]. Considering that the evaluation of inflammation in the small intestine was carried out by various diagnostic methods, we accepted: L1, lesions located in 1/3 distal part of the ileum; L2, lesions located in the colon; L3, lesions located in the ileum and colon. Due to the small number of patients with both penetrating (B3) or stricturing and penetrating either at the same or different times (B2B3) phenotype of CD, they were assigned to one group (B3+B2B3). Perianal CD was defined as fistulas, abscesses, fissures or condylomas with a minimum length of 1 cm. A child with growth delay was defined as a child whose height was less than the value of the 3rd percentile for age and sex on the percentile charts (which corresponds to the value <−1.88 SD) or when the height curve changed its position on the percentile charts by at least 2 percentile channels. Percentile charts were used in accordance with the national standards for gender and age. The main characteristics of the study and control group are shown in Table 1. Laboratory tests routinely performed include complete blood count, inflammatory markers (serum CRP, ESR, AAG and PCT), cholesterol level and serum albumin. FC, CRP and ESR were measured in all patients with CD, AAG in 36/54 patients and PCT in 29/54 children. Serum inflammatory markers were determined in the Central Laboratory of Clinical Hospital No 1 in Wroclaw according to standard methods. Stool samples for assessment of FC concentration were collected in plastic tubes without the addition of stabilizers. The samples were immediately frozen at −20 °C. FC was measured using an ELISA kit (Buehlmann Laboratories AG, Schönenbuch, Switzerland). According to the manufacturer’s instructions, the test is useful in the differential diagnosis of functional and organic diseases of the digestive tract, with sensitivity of 84.4% and specificity of 94.5%. Manufacturers of test kits recommend a cut-off FC concentration of 50 µg/g, above which results are positive. FC in the range of 50–200 µg/g suggests intestinal mucosal inflammation of low intensity, and over 200 µg/g suggests an organic gastrointestinal tract disease. Clinical activity of CD was determined according to PCDAI, which classifies children into 4 disease activity categories: clinical remission (0–10 points), mild disease (11–25 points), moderate disease (26–50 points) and severe disease (>50 points) [11]. In our study, clinical remission was found in 17 patients (31.48%), mild CD in 18 patients (33.33%), moderate CD in 18 patients (33.33%) and severe CD in 1 patient (1.85%). Taking into account that only 1 patient had a severe course, patients with moderate to severe CD were assigned to one group. 

The endoscopic activity of CD was determined according to SES-CD [12]. Due to the different interpretation criteria of SES-CD in various clinical trials [13], in our study, results were analyzed in relation to three types of criteria for endoscopic activity: (1) endoscopic remission: 0–3 points, mild, moderate or severe form: >3 points; (2) endoscopic remission: 0–2 points, mild form: 3–6 points, moderate form: 7–15 points, severe form: > 15 points; (3) endoscopic remission: 0–3 points, mild form: 4–10 points, moderate form: 11–19 points, severe form: >19 points.

## 4. Statistical Analysis

Statistical analysis was performed using Statistica PL 10 (StatSoft Inc., Tulsa, OK, USA). Baseline characteristics were presented as the mean, median, standard deviation and range (minimum and maximum). Variable distributions were tested with the Kolmogorov–Smirnov test of normality. To compare clinical indices between the two groups, the Student’s *t*-test was performed, and if the normality test failed, the exact Mann–Whitney test was used. The quantitative variables were described using one-way analysis of variance (ANOVA). The correlation of two quantitative variables was assessed by a Pearson correlation coefficient.

## 5. Results 

The mean FC, CRP, ESR and AAG values were significantly higher in patients with CD compared with controls (all *p* < 0.0001), whereas mean serum PCT levels were comparable in both groups (Table 2). We found a trend of higher median levels of FC, CRP and AAG in patients with CD with extensive lesions in the ileum and colon (L3-1013.9 µg/g, 11.7 mg/L, 1.55 g/L) compared to patients with lesions located only in the colon (L2- 927.98 µg/g, 10.25 mg/L, 1.24 g/L) or in the ileum (L1- 897.36 µg/g, 6.45 mg/L, 1.4 g/L), but the differences were not statistically significant (*p* = 0.579, *p* = 0.669, *p* = 0.84). Additionally, no association between inflammatory markers and CD behavior (B1, B2, B3+B2B3) was found (Table 3). A statistically significant (*p* <0.05) positive correlation was found between clinical activity according to PCDAI and FC (r = 0.338; *p* = 0.012), CRP (r = 0.43; *p* = 0.001), ESR (r = 0.384; *p* = 0.004) and AAG (r = 0.45; *p* = 0.004). No correlation between clinical activity and PCT was found (Table 4, Figure 1).

Analyzing whether the tested inflammatory markers could be used to distinguish different grades of CD clinical activity according to PCDAI, we found that FC was higher in patients with moderate to severe disease than in children with clinical inactive or mild disease, but the differences were not statistically significant (*p* = 0.14). Furthermore, it was shown that CRP, ESR and AAG were significantly higher in children with moderate to severe disease than in patients with clinical remission. Additionally, ESR and AAG were significantly higher in children with mild disease than in patients with clinical remission (Table 5). CD endoscopic activity according to SES-CD significantly positively correlated with FC (r = 0.428; *p* = 0.001), CRP (r = 0.27; *p* = 0.048) and AAG (r = 0.337; *p* = 0.036), but not with ESR and PCT (r = 0.19; *p* = 0.94) (Table 4, Figure 2). Furthermore, in patients with active mucosal inflammation (SES-CD, > 3 points) FC, serum CRP and AAG were significantly higher (*p* = 0.01, *p* = 0.047; *p* = 0.012) compared with patients with endoscopic remission (SES-CD, 0–3 points). If the endoscopic findings were graded as mucosal remission, 0–2 points SES-CD, mild disease: 3–6 points SES-CD, moderate disease 11–15 points SES-CD, severe disease > 15 points SES-CD, no statistically significant differences levels of tested inflammatory markers were found in compared groups. If the endoscopic findings were graded as mucosal remission, 0–3 points SES-CD, mild disease 4–10 points SES-CD, moderate disease 11–19 points SES-CD, severe disease > 19 points, FC was significantly higher in patients with severe endoscopic activity than in children with inactive mucosal disease (*p* = 0.038). 

We also found that among the tested inflammatory markers, FC was the only one that was significantly higher in children with perianal CD compared with patients without perianal disease (median = 1072.36 µg/g; median = 798.36 µg/g; *p* = 0.039).

To determine the usefulness of biomarkers in the diagnosis of CD in children, we also performed ROC analysis (area under the curve, AUC) of the inflammatory markers: FC, CRP, ESR, AAG and PCT. The AUC for FC was 0.9504. The sensitivity of the method to differentiate patients with CD from non-inflammatory gastrointestinal diseases was 88.89%, and specificity was 94.33% with the cut-off value of 248.97 µg/g. Detailed data on the ROC analysis for all assessed biomarkers are presented in Figure 3 and Table 6.

## 6. Discussion

The results of the present study confirm that the determination of the FC level is useful in the differential diagnosis of CD and non-inflammatory gastrointestinal tract diseases in children, which is in agreement with the results of previous studies [14,15,16]. This is particularly important because of the limited number of reports of children with inflammatory bowel diseases (IBD). A meta-analysis by Van Rheenen et al. [17] demonstrated that the initial differential diagnosis based on determining the concentration of FC in children with suspected IBD resulted in a 35% decrease in the number of performed gastrointestinal endoscopies. Thus, assessment of the FC concentration in everyday gastroenterology practice significantly reduces the number of invasive procedures performed in the differential diagnosis of IBD, which is especially important in the pediatric population. We also found that widely used inflammatory markers (CRP, ESR and AAG) might be useful for differential diagnosis of CD and inorganic gastrointestinal tract diseases, which is in line with previous studies [18,19]. However, a significant number of children with CD may not have elevated inflammatory markers levels even in the setting of active disease. In the study by Sładek et al. [19], 38% of children with newly diagnosed CD had normal ESR values, and 40% had normal CRP levels. Additionally, in the study by Buderus et al. [20], in 10% of children with active CD, the CRP level was within normal limits. This may delay the diagnosis or may lead to the unnoticed progression of CD in children [1,3]. Elevated levels of inflammatory markers may also be observed in many other diseases, particularly infections. However, due to the low sensitivity and specificity of these inflammatory markers for distinguishing CD from other gastrointestinal tract diseases or for the gradation of CD activity, the clinical usefulness of their isolated testing is limited [14,19,21]. Taken together, the assessment of clinical symptoms and combined use of blood-based non-invasive inflammatory biomarkers may be useful in the differential diagnosis of IBD and inorganic gastrointestinal disorders in children [22,23].

Currently, non-invasive assessment of clinical activity CD in children PCDAI is used [24]. This study demonstrates that FC is positively correlated with disease activity according to PCDAI, which is in accordance with the results of other studies [16,25,26]. This study also demonstrates that FC does not differentiate categories of clinical activity according to PCDAI into inactive, mild, moderate or severe CD in children. These results are in contrast with some previous studies [16,26]. This could be due to the small number of patients in the compared subgroups in this study. It is also increasingly noted that the commonly used indexes for assessing the activity of CD in adults and children (PCDAI) do not correlate with endoscopic activity, and they are unreliable for endoscopic disease severity assessment, especially in patients with poor clinical symptoms. The biomarker that seems to correlate well with the activity of mucosal lesions, even in clinically asymptomatic patients, is FC [27,28]. It has been shown that patients with clinically inactive CD, according to PCDAI, may have increased FC levels despite previous therapy, which results directly from ongoing mucosal inflammation [28]. The differences in the results compared with other studies can be explained by the fact that a large number of children with CD received treatment before entering our study. Moreover, PCDAI includes subjective patient reporting components such as general well-being or abdominal pain. The subjectivity of those references limits the ability to the true measurement of the clinical activity. Additionally, the role of FC as an indicator of clinical activity in patients with CD was not ultimately evaluated. There is still a need for further prospective studies with a large number of patients in remission and with mild, moderate or severe CD in children. It is necessary to finally assess the importance of FC in the diagnostics and monitoring of CD in children considering age-dependent modifications, lesions location and the type of therapy used. Additionally, used in daily practice, serum inflammatory markers (CRP, ESR and AAG) positively correlate with PCDAI, so they are useful in evaluating the clinical activity of CD in children [29,30]. The results of our study confirm this correlation. In our study, we also observed that based on serum AAG and ESR, there is a possibility to distinguish between patients with clinical remission (≤10 points, PCDAI) and active disease (>10 points, PCDAI). CRP is useful in differentiation between clinical remission (≤10 points, PCDAI) and moderate to severe disease (≥26 points, PCDAI). These observations underline that the determination of these serum non-invasive inflammatory markers should not be abandoned, although considering the limitations resulting from low sensitivity and specificity in diagnostics and monitoring of CD in children [31]. However, it is necessary to interpret the results taking into account other conditions that could increase levels of serum inflammatory markers. 

Currently, the only way to adequately assess the severity of inflammatory mucosal lesions is by performing gastrointestinal endoscopy [2,4,28,32]. There is a growing need to replace this invasive diagnostic method with a non-invasive marker, at least with an equal sensitivity, such as gastrointestinal endoscopy in the assessment of mucosal healing and controlling the effectiveness of therapy in children with CD [4,28,32]. Several studies have shown that FC, CRP and AAG correlated with endoscopic activity according to SES-CD. Our results also support previous studies [22,28,33,34]. Evaluating how exactly inflammatory markers should be used in clinical practice requires further analysis. Zubin et al. [28] showed that CRP correlated positively with the severity of mucosal inflammation according to SES-CD at diagnosis in children with CD, but there was no such relationship after induction therapy. Schoepfer et al. [22] showed a similar correlation between FC, CRP and leukocytosis and SES-CD in adults, but the correlation of FC was stronger than other inflammatory markers. Moreover, FC was the only marker that could discriminate between endoscopic active and inactive disease. Szczepański et al. [35] and Aomatsu et al. [33] showed that in children with CD, FC is a good marker of mucosal healing, even if endoscopic remission is defined only as 0 points according to SES-CD. Similarly, the present study showed that in children with active endoscopic CD (>3 points, SES-CD), FC levels were significantly higher than in patients with endoscopic remission (0–3 points, SES-CD). In our study, we also observed a positive correlation between endoscopic activity assessed by SES-CD and CRP and AAG. In the literature, there are only a few studies, with no consistent results, on the value of correlation between selected inflammatory markers and endoscopic activity according to SES-CD in children. Zubin et al. [28] studied the correlations between various activity indicators (FC, CRP, PCDAI) and the severity of mucosal lesions in children with CD, and they showed that during maintenance therapy, only FC levels correlated with endoscopic activity. A variety of results from previous studies confirm that only using a combination of symptoms, clinical observations and the determination of multiple biomarkers, especially FC and CRP, provides the possibility of non-invasive identification of children with suspected CD and also to evaluate the effectiveness of therapy in patients with already diagnosed CD [23,28]. The overall number of studies, especially in the pediatric population, that have evaluated serum and fecal inflammatory markers and their relationship with endoscopic disease activity is small. Moreover, comparing the results of those is complicated further because of the assumption of various SES-CD cut-off points [13,22,36]. In this study, we tested the usefulness of non-invasive biomarkers in distinguishing different grades of the endoscopic activity assessed by SES-CD according to various cut-off points. Similar to Schoepfer et al. [22], we found that FC can be useful in differentiation between patients with endoscopic remission (0–3 points, SES-CD) from those with severe endoscopic activity (>19 points, SES-CD). Moreover, this contrasts with the results presented by Langhorst et al. [18] from adult patients where none of evaluated inflammatory markers (FC, fecal lactoferrin, CRP) were useful in distinguishing grades of endoscopic activity according to SES-CD assuming the same cut-off points. Based on these observations and due to the small numbers of patients in the studied subgroups, it is difficult to present a final conclusion.

Another important issue in the daily care of children with CD is personalized management according to, not just disease activity and severity, but also to the location of inflammatory lesions (systemic or locally therapy?; if locally rectal infusions, suppositories or mousses?) (2). The best way to determine the location of inflammatory lesions would be to use non-invasive inflammatory markers. So far, only a few studies have evaluated the relationship between raised non-invasive inflammatory markers and the location of lesions in children with CD, and the results are conflicting [13,25,37]. It has been shown that FC correlates with lesions located in the lower part of the gastrointestinal tract [25] and with isolated changes in the small intestine [37]. In this present study, no correlation between tested non-invasive markers and disease location was observed. Our results are in agreement with previous studies [8,14,19]. In the present study, no correlation between inflammatory markers and CD behavior was shown, which is similar to the results presented by Schoepfer et al. [22] from adult patients. No studies from the pediatric population were found.

Previously published studies, especially from adult patients, have shown that PCT is not useful for the diagnosis of IBD and assessment of disease behavior, activity and location [18,38,39]. Our results are in agreement with those studies.

## 7. Summary

Determination of FC, CRP, ESR and AAG is useful in the differential diagnosis of CD and non-inflammatory gastrointestinal tract diseases in children. Because of the relatively high sensitivity, the most important biomarker seems to be FC. Due to the lack of specificity in gastrointestinal tract disease diagnosis, the usefulness of other tested biomarkers is limited. All tested non-invasive biomarkers, except PCT, correlated with the clinical and endoscopic activity in children with CD. However, based on the non-invasive biomarkers, gradation of CD clinical activity according to PCDAI and endoscopic activity according to SES-CD, determination of lesions location and disease behavior should not be performed. In this study, we confirmed the usefulness of the daily practice of the tested non-invasive inflammatory markers, except PCT, in identifying children with suspected CD from the pediatric population of patients with gastrointestinal symptoms, as well as in monitoring the CD course. Due to the lack of accuracy in distinguishing exact grades of clinical and endoscopic activity of CD in children, which determines the location and disease behavior of lesions, there is still a need to look for another biomarker that would reduce the number of invasive diagnostic procedures performed.

## 8. Conclusions

The determination of FC, CRP, ESR and AAG was useful in the differential diagnosis of CD and non-inflammatory gastrointestinal tract diseases in children;FC, CRP, ESR and AAG positively correlated with the clinical activity of CD in children assessed by PCDAI;The determination of FC, CRP and AAG was useful in the differentiation between active and inactive endoscopic changes in CD in children according to SES-CD;The determination of inflammatory markers was not useful in evaluating the location and phenotype of CD lesions in children.

## Figures and Tables

**Figure 1 jcm-11-06086-f001:**
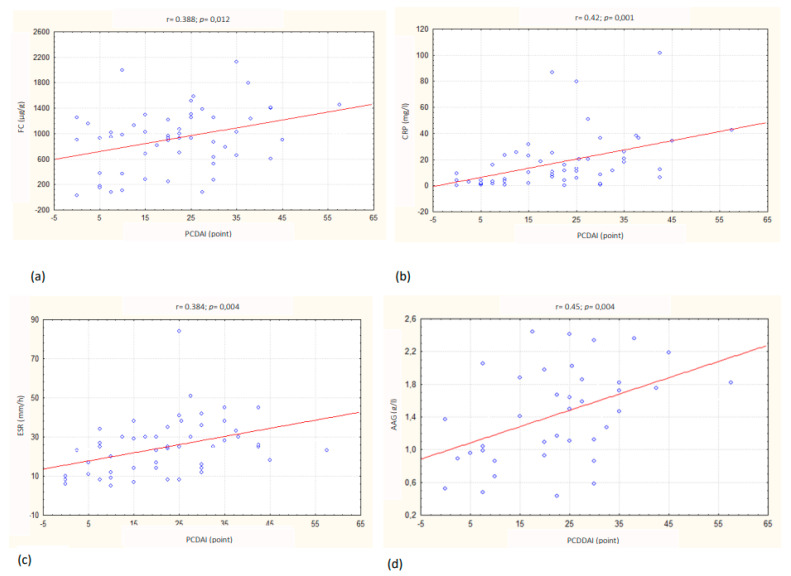
Correlation between (**a**) fecal calprotectin (FC), (**b**) C-reactive protein (CRP), (**c**) erythrocyte sedimentation rate (ESR), (**d**) seromucoid (AAG) and clinical activity according to the Pediatric Crohn’s Disease Activity Index (PCDAI) in children with Crohn’s disease.

**Figure 2 jcm-11-06086-f002:**
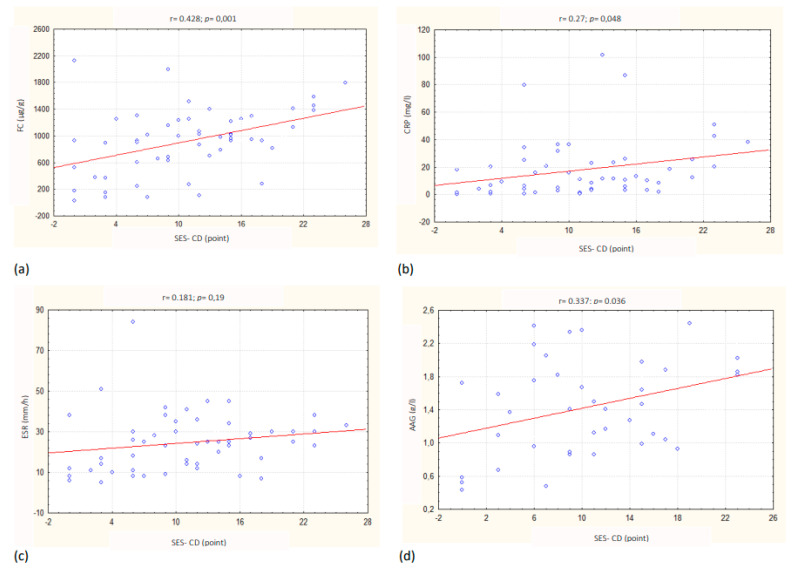
Correlation between (**a**) fecal calprotectin (FC), (**b**) C-reactive protein (CRP), (**c**) erythrocyte sedimentation rate (ESR), (**d**) seromucoid (AAG) and endoscopic activity according to the Simple Endoscopic Score for Crohn’s Disease (SES-CD) in children with Crohn’s disease.

**Figure 3 jcm-11-06086-f003:**
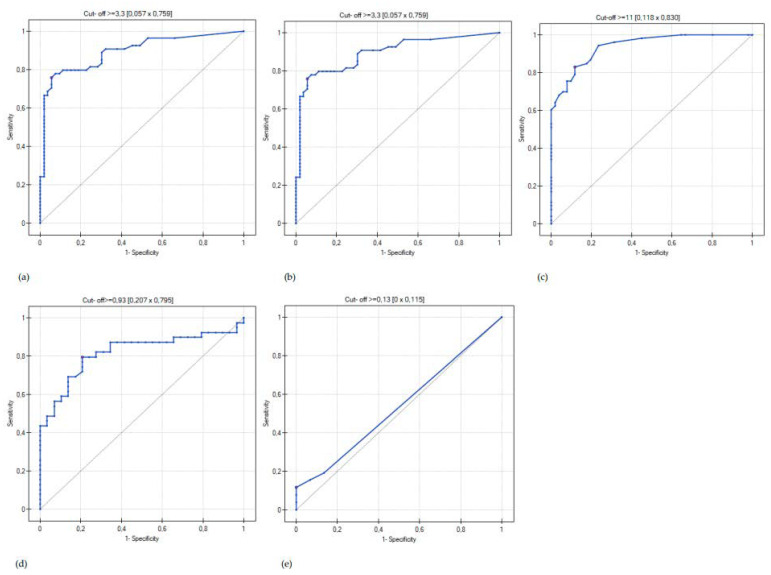
Receiver operator characteristic (ROC) curve for serum and fecal biomarkers: fecal calprotectin (FC) (**a**), C-reactive protein (CRP) (**b**), erythrocyte sedimentation rate (ESR) (**c**), seromucoid (AAG) (**d**) and procalcitonin (PCT) (**e**) in the differentiation of Crohn’s disease in children with non-inflammatory diseases of the gastrointestinal tract.

**Table 1 jcm-11-06086-t001:** Characteristics of study and control groups.

	Crohn’s Disease (*n* = 54)	Control Group (*n* = 47)
Gender: *n* (%)	male: 30 (55.56%) female: 24 (44.44%)	male: 30 (63.83%) female: 17 (36.1%)
Age [years]: median (range)	14.83 (5.58–18.00)	13.94 (3.75–17.66)
Weight [kg]: median (range)	44.7 (16.1–96.8)	49.5 (13.6–116.6)
Growth [cm]: median (range)	157.25 (109.0–191.0)	160.0 (104.0–188.0 )
Disease location: *n* (%) - L1 (distal 1/3 ileum ± limited cecal disease) - L2 (colonic) - L3 (ileocolonic) - L4 (upper disease; L4a- proximal to the ligament of Treitz; L4b- distal to the ligament of Treitz and proximal to distal 1/3 ileum)	- 6 (11.11%) - 24 (44.44%) - 23 (42.59%) - 1 (1.85%)	-
Disease behavior: *n* (%) - B1 (non-stricturing, non-penetrating) - B2 (structuring) - B3 (penetrating) + B2B3 (both penetrating and stricturing disease; either at the same or different times)	- 33 (61.11%) - 12 (22.22%) - 9 (16.66%)	-
p: perianal disease modifier	30 (44.44%)	-
Growth - G0 (no evidence of growth delay) - G1 (growth delay)	- 42 (77.78%) - 12 (22.22%)	-

**Table 2 jcm-11-06086-t002:** Comparison of inflammatory markers in children with Crohn’s disease and control group.

Group	Crohn’s Disease					Control Group					*p*-Value; *p*-Value (Mann–Whitney Test)
	*n*	Mean	± SD	Range	Median	*n*	Mean	± SD	Range	Median	
FC ^1^ (µg/g)	54	920.12	487.15	29.4–2126.47	936.85	47	60.24	77.96	1.9–397.74	33.18	<0.0001; <0.0001
CRP ^2^ (mg/L)	54	17.72	21.61	0.2–101.7	10.75	47	1.36	3.67	0.2–25.1	0.3	<0.0001; <0.0001
ESR ^3^ (mm/h)	54	24.64	14.3	5–84	25	46	6.09	4.6	2–19	5	<0.0001; <0.0001
AAG ^4^ (g/L)	39	1.42	0.57	0.43–2.44	1.41	30	0.79	0.24	0.47–1.42	0.74	<0.0001; <0.0001
PCT ^5^ (ng/mL)	26	0.076	0.093	0.05–0.52	0.05	17	0.052	0.007	0.05–0.08	0.05	0.3; 0.45

^1^ FC—fecal calprotectin; ^2^ CRP—C-reactive protein; ^3^ ESR—erythrocyte sedimentation rate; ^4^ AAG—seromucoid, ^5^ PCT—procalcitonin.

**Table 3 jcm-11-06086-t003:** Comparison of fecal calprotectin (FC) and serum concentration C-reactive protein (CRP), erythrocyte sedimentation rate (ESR), seromucoid (AAG) and procalcitonin (PCT) depending on the location of the endoscopic lesions of Crohn’s disease (CD) in children.

	FC (µg/g)		CRP (mg/L)		ESR (mm/h)		AAG (g/L)		PCT (ng/mL)	
	*n*	Median (Min–Max)	*n*	Median (Min–Max)	*n*	Median (Min–Max)	*n*	Median (Min–Max)	*n*	Median (Min–Max)
Endoscopic localization of lesions in CD										
- L1	- 6	- 897.36 (248.97–2126.47)	- 6	- 6.45 (4–25.2)	- 6	- 20 (8–38)	- 4	- 1.4 (0.86–1.75)	- 2	- 0.05 (0.05–0.05)
- L2	- 24	- 927.98 (78.46–1585.9	- 24	- 10.25 (0.2–101.7)	- 24	- 25 (5–84)	- 18	- 1.24 (0.43–2.44)	- 13	- 0.05 (0.05–0.52)
- L3	- 23	- 1013.9 (79.4–1794.2)	- 23	- 11.7 (1.3–86.7)	- 23	- 25 (7–45)	- 16	- 1.55 (0.48–2.19)	- 10	- 0.05 (0.05–0.14)

ANOVA *p*-value		*p* = 0.579		*p* = 0.669		*p* = 0.684		*p* = 0.84		*p* = 0.69
Clinical characteristics of patients with CD										
- B1	- 33	- 820.75 (29.4–2126.47)	- 33	- 6.8 (0.2–101.7)	- 33	- 22 (5–84)	- 22	- 1.11 (1.43–2.41)	- 17	- 0.05 (0.05–0.52)
- B2	- 12	- 1114.72 (685.61–1794.2)	- 12	- 15.9 (2.7–86.7)	- 12	- 24 (8–38)	- 9	- 1.41 (0.89–2.19)	- 5	- 0.05 (0.05–0.14)
- B3 + B2B3	- 9	- 1025.03 (270.8–1385.7)	- 9	- 11 (5.7–50.9)	- 9	- 25 (14–41)	- 8	- 1.75 (1.41–2.05)	- 4	- 0.05 (0.05–0.05)

ANOVA *p*-value		*p* = 0.157		*p* = 0.46		*p* = 0.96		*p* = 0.14		*p* = 0.82

**Table 4 jcm-11-06086-t004:** Correlation between the clinical activity of Crohn’s disease (CD) in children assessed using the Pediatric Crohn’s Disease Activity (PCDAI) scale and the simplified scale of endoscopic changes and their severity in Crohn’s disease (Simple Endoscopic Score for Crohn’s disease, SES-CD) and fecal calprotectin and the concentration of serum C-reactive protein (CRP), erythrocytes sedimentation rate (ESR), seromucoid (AAG) and procalcitonin (PCT).

	FC (µg/g); *n* = 54		CRP (mg/L); *n* = 54		ESR (mm/h); *n* = 54		AAG (g/L); *n* = 39		PCT (ng/mL); *n* = 26	
Pearson Correlation Coefficient—r *p*-Value	r	*p*	R	*p*	r	*p*	r	*p*	r	*p*
PCDAI	0.338	0.012	0.43	0.001	0.384	0.004	0.45	0.004	0.12	0.56
SES-CD	0.428	0.001	0.27	0.048	0.181	0.19	0.337	0.036	0.016	0.94

**Table 5 jcm-11-06086-t005:** Comparison of fecal calprotectin (FC) and C-reactive protein (CRP), erythrocytes sedimentation rate (ESR), seromucoid (AAG) and procalcitonin (PCT) in the blood serum depending on the clinical activity of Crohn’s disease (CD) in children assessed using the Pediatric Crohn’s Disease Activity Index (PCDAI) scale and the Simple Endoscopic Score for Crohn’s disease (SES-CD).

	FC (µg/g)		CRP (mg/L)		ESR (mm/h)		AAG (g/L)		PCT (ng/mL)	
Activity of CD Clinical- PCDAI/Endoscopic- SES-CD	*n*	Median (Min–Max)	*n*	Median (Min–Max)	*n*	Median (Min–Max)	*n*	Median (Min–Max)	*n*	Median (Min–Max)
PCDAI (points)										
- 0–10—clinical remission (G1)	- 16	- 716.99 (29.4–2000)	- 16	- 5.01 (0.2–101.7)	- 16	- 15.07 (5–34)	- 10	- 0.98 (0.48–2.5)	- 6	- 0.05 (0.05–0.05)
- 11–25—mild disease (G2)	- 18	- 989.21 (248.97–1297.23)	- 20	- 20.5 (0.7–50.9)	- 20	- 27.15 (5–45)	- 15	- 1.54 (0.4–0.44)	- 10	- 0.07 (0.05–0.14)
- 26–50 + > 50—moderate and severe disease (G3)	- 20	- 1023.93 (78.482126.47)	- 18	- 25.93 (0.7–86.7)	- 18	- 29.83 (12–51)	- 14	- 1.63 (0.58–0.36)	- 10	- 0.05 (0.05–0.52)
ANOVA *p*-value		*p* = 0.14		*p* = 0.012; *p*_1_ = 0.083;		*p* = 0.006; *p*_1_ = 0.034;		*p* = 0.012; *p*_1_ = 0.044;		*p* = 0.549
*p*; *p*_1_—G1:G2; *p*_2_—G1:G3; *p*_3_—G2:G3				*p*_2_ = 0.015; *p*_3_ = 0.711		*p*_2_ = 0.009; *p*_3_ = 0.821		*p*_2_ = 0.019; *p*_3_ = 0.907		
I. SES-CD (points)										
- 0–3—(endoscopic remision)	- 10	- 566.65 (29.4–2126.47)	- 10	- 16 (0.2–20.4)	- 10	- 18 (5–51)	- 7	- 0.94 (0.43–1.72)	- 3	- 0.05 (0.05–0.05)
- >3—(mild, moderate or severe lesions)	- 44	- 1000.46 (79.4–2000)	- 44	- 22.41 (0.4–101.7)	- 44	- 26 (7–84)	- 32	- 1.53 (0.48–2.44)	- 23	- 0.08 (0.05–0.52)
*p*-value		*p* = 0.01		*p* = 0.047		*p* = 0.127		*p* = 0.012		*p* = 0.617
II. SES-CD (points)										
- 0–2 pkt. (endoscopic remision)	- 6	- 694.95 (29.4–928.9)	- 10	- 5.51 (0.2–20.4)	- 10	- 18 (5–51)	- 4	- 0.81 (0.43–1.72)	- 3	- 0.05 (0.05–0.05)
- 3–6 pkt. (mild lesions)	- 11	- 693.95 (78.46–1305.4)	- 16	- 20.31 (0.7–79.7)	- 16	- 26.65 (8–84)	- 8	- 1.5 (0.67–2.41)	- 3	- 0.077 (0.05–0.13)
- 7–15 pkt. (moderate lesions)	- 25	- 420.2 (79.4–2000)	- 22	- 17.59 (0.4–101.7)	- 22	- 24.55 (7–45)	- 19	- 1.44 (0.48–2.36)	- 13	- 0.09 (0.05–0.52)
- >15 pkt. (severe lesions)	- 12	- 404.6 (279.56–1794.2)	- 6	- 31.61 (12.4–50.9)	- 6	- 29.83 (23–38)	- 8	- 1.64 (0.93–2.44)	- 6	- 0.055 (0.05–0.08)
ANOVA *p*-value		*p* = 0.055		*p* = 0.113		*p* = 0.397		*p* = 0.117		*p* = 0.401
III. SES-CD										
- 0–3 pkt. (endoscopic remision)	- 10	- 566.65 (29.4–2126.47)	- 10	- 5.51 (0.2–20.4)	- 10	- 18 (5–51)	- 7	- 0.94 (0.43–1.72)	- 3	- 0.05 (0.05–0.05)
- 4–10 pkt. (mild lesions)	- 16	- 912.48 (79.4–2000)	- 16	- 20.31 (0.7–79.7)	- 16	- 8 (8–84)	- 14	- 1.61 (0.48–2.41)	- 9	- 0.07 (0.05–0.14)
- 11–19 pkt.(moderate lesions)	- 22	- 938.99 (104.71–1517.2)	- 22	- 17.59 (0.4–101.7)	- 22	- 24.55 (7–45)	- 15	- 1.39 (0.86–2.44)	- 11	- 0.09 (0.05–0.52)
- >19 (severe lesions)	- 6	- 1460.46 (1128.8–1794.2)	- 6	- 31.61 (12.4–50.9)	- 6	- 29.83 (23–38)	- 3	- 1.9 (1.82–2.02)	- 3	- 0.06 (0.05–0.08)
ANOVA *p*-value		*p* = 0.038		*p* = 0.113		*p* = 0.397		*p* = 0.029		*p* = 0.886

**Table 6 jcm-11-06086-t006:** AUC analysis and sensitivity and specificity of fecal and serum biomarker values: fecal calprotectin (FC) C-reactive protein (CRP), erythrocyte sedimentation rate (ESR), seromucoid (AAG) and procalcitonin (PCT) in the diagnosis of Crohn’s Disease in children.

	AUC	95% IC	*p*	Cut-Off	Sensitivity	Specificity
FC (µg/g)	0.95	(0.91–0.99)	<0.001	248.97	0.89	0.94
CRP (mg/L)	0.89	(0.83–0.96)	<0.001	3.3	0.76	0.94
ESR (mm/h)	0.94	(0.89–0.97)	<0.001	11	0.83	0.88
AAG (g/L)	0.81	(0.71–0.92)	<0.001	0.93	0.79	0.84
PCT (ng/mL)	0.54	(0.42–0.65)	0.69	0.13	0.11	1

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
