# Peer review of "Evaluation of Fecal Calprotectin, Serum C-Reactive Protein, Erythrocyte Sedimentation Rate, Seromucoid and Procalcitonin in the Diagnostics and Monitoring of Crohn’s Disease in Children"

_jcm, 2022, doi:10.3390/jcm11206086_

Round 1

Reviewer 1 Report

1)In line 40, please after Currently adding (,).

2)Some lines do not have references . please check

3) In Table 1, please explain your means of (Growth) in detail.

4) your paper doesn't have novelty. For many years it has been determined that FC is the main factor in IBD care. Why do you follow your patients and compare the FC level before and after treatment or flare-up and remission? And also, why only check FC  among CD patients ? UC patients are as important as a CD. If you can add data on remission status and UC patients help to increase the novelty of your study.

Author Response

Dear Reviewer,

Thank you for reading and thorough analysis of our report as well as valuable comments.

The work has been corrected in accordance with the comments contained in points 1-3 (marked in yellow in the text).

Responding to point 4, we would like to emphasize that indeed this study confirms previous reports on the importance of calprotectin as a non-invasive biomarker of inflammatory activity in the intestine in patients with Crohn's disease, but also summarizes the correlation of various widely available markers once considered important for the assessment of disease activity. as well as both clinical (PCDAI) and endoscopic (SES-CD) rating scales. Therefore, the paper can provide a good summary for the reader that the results of patient studies or different scales of Crohn's disease activity should be interpreted with the greatest caution.

In this study, comparing patients treated in different ways, with different clinical and endoscopic activity, in the period of exacerbation and remission, also emphasizes the possibility of the assessed markers and scales to assess the effectiveness of broadly understood therapeutic treatment.

The aim of this study was to evaluate calprotectin and other inflammatory markers in pediatric patients with Crohn's disease. Similar data, which only concerned patients with ulcerative colitis, have already been compiled and the results have been published.

We hope that You will be satisfied with the changes we have made.

Regards,

Katarzyna Akutko

Reviewer 2 Report

In this non-randomized study, the Authors aim to evaluate the accuracy of a few non-invasive biomarkers in differential diagnosis and the phenotype among patients with Crohn’s disease (CD) The authors showed that Faecal calprotectin, C-reactive protein, erythrocyte sedimentation rate and seromucoid are useful in differentiation CD from non-inflammatory gastrointestinal tract diseases but not in evaluating activity and phenotype of the disease.

The study was overall well-conducted, and the conclusions are robust and consistent with the shown data.
I have the following critiques and recommendations:

1.     Material and methods: It seems to be a cross-sectional study. But it is extremely important that the authors provide more information concerning the study design and the patient’s enrollment (they included all the patients followed in their unit? and They use a specific time interval for the enrollment? Was It prospective or retrospective?)

2.     The Authors declare that “all children with CD were examined with 73 abdomen ultrasounds, the bowel wall thickness was measured”. Evaluating the correlation between wall thickness (expressed as a dichotomous variable) could be a valuable additional analysis. 

Minor revision

-       The italics in lines 71-75 seem to be unnecessary  

Author Response

Dear Reviewer,

thank you very much for your opinion.

Referring to your tips:

- "Material and methods" section has been modified and supplemented

- The  statement about USG of the abdomen was intended to emphasize that the diagnosis of Crohn's disease in patients enrolled in the study was made according to Porto criteria, including imaging techniques such as abdominal ultrasound and MRI enterography during the diagnostic process. As not all patients had abdominal ultrasound during hospitalization, during which non-invasive markers of inflammation in blood and feces were measured, this parameter was not included in the analyzes, because only a simultaneous measurement and then the comparison of these elements would be a real added value . In order not to enter redundant information, the statement "the bowel wall thickness was measured" has been deleted.

We hope that you will be satisfied with the changes we have made.

Regards,

Katarzyna Akutko